# Landscape Representation by a Permanent Forest Plot and Alternative Plot Designs in a Typhoon Hotspot, Fushan, Taiwan

Jonathan Peereman [1,2], James Aaron Hogan [3] and Teng-Chiu Lin [2,*]

[1] Biodiversity Program, Taiwan International Graduate Program, Biodiversity Research Center, Academia Sinica and National Taiwan Normal University, Nankang District, Taipei 11529, Taiwan; 80650007s@ntnu.edu.tw
[2] Department of Life Science, National Taiwan Normal University, No. 88, Section 4, TingChow Road, Wenshan District, Taipei 11677, Taiwan
[3] Department of Biological Sciences, Florida International University, Miami, FL 33199, USA; jhogan@fiu.edu
[*] Correspondence: tclin@ntnu.edu.tw

**Abstract:** Permanent forest dynamics plots have provided valuable insights into many aspects of forest ecology. The evaluation of their representativeness within the landscape is necessary to understanding the limitations of findings from permanent plots at larger spatial scales. Studies on the representativeness of forest plots with respect to landscape heterogeneity and disturbance effect have already been carried out, but knowledge of how multiple disturbances affect plot representativeness is lacking—particularly in sites where several disturbances can occur between forest plot censuses. This study explores the effects of five typhoon disturbances on the Fushan Forest Dynamics Plot (FFDP) and its surrounding landscape, the Fushan Experimental Forest (FEF), in Taiwan where typhoons occur annually. The representativeness of the FFDP for the FEF was studied using four topographical variables derived from a digital elevation model and two vegetation indices (VIs), Normalized Difference Vegetation Index (NDVI) and Normalized Difference Infrared Index (NDII), calculated from Landsat-5 TM, Landsat-7 ETM+, and Landsat-8 OLI data. Representativeness of four alternative plot designs were tested by dividing the FFDP into subplots over wider elevational ranges. Results showed that the FFDP neither represents landscape elevational range (<10%) nor vegetation cover (<7% of the interquartile range, IQR). Although disturbance effects (i.e., ΔVIs) were also different between the FFDP and the FEF, comparisons showed no under- or over-exposure to typhoon damage frequency or intensity within the FFDP. In addition, the ΔVIs were of the same magnitudes in the plots and the reserve, and the plot covered 30% to 75.9% of IQRs of the reserve ΔVIs. Unexpectedly, the alternative plot designs did not lead to increased representation of damage for 3 out of the 4 tested typhoons and they did not suggest higher representativeness of rectangular vs. square plots. Based on the comparison of mean Euclidian distances, two rectangular plots had smaller distances than four square or four rectangular plots of the same area. Therefore, this study suggests that the current FFDP provides a better representation of its landscape disturbances than alternatives, which contained wider topographical variation and would be more difficult to conduct ground surveys. However, upscaling needs to be done with caution as, in the case of the FEF, plot representativeness varied among typhoons.

**Keywords:** forest disturbances; typhoon; forest permanent plots; representativeness; landscape ecology

## 1. Introduction

Forest studies commonly use permanent forest dynamics plots, which may serve as reference sites for the larger studied environment. Their uses range from a single vegetation census to multiple censuses spanning decades, as is the case in long-term forest research such as those of the Smithsonian's Forest-GEO network [1,2]. In long-term research, permanent plots (PP) can be large continuous blocks (e.g., Forest-GEO plots) or consist of several smaller plots dispersed over the landscape (e.g., RAINFOR plots, [3]). The PPs permit important discoveries in plant demography, community dynamics, to ecosystem change ecology [4–10] (among numerous others), made possible by standardized regular censuses that require a tremendous amount of time and effort. As a result, PP design is the product of trade-offs between the ecological questions being asked, and practical constraints of regularly censusing them, such as accessibility and plot size. However, researchers must consider the limitations inherent to the PP design if PP-based findings are to be scaled up to the landscape and regional scales, uses that may be beyond initial research goals.

Studying the representativeness of PPs is necessary to verify if observations from them can represent their environments at the landscape scale, and also to know which parts of the landscape remain under-represented [11,12]. For example, a study comparing plots and landscape through LiDAR imagery and image spectrometry in the Amazon found biases for forest biomass and canopy height in both lowland and montane forests [13]. A similar study in Hawaii revealed that PPs underestimated the variation in forest height and had a slight bias for taller forests [14]. PPs provide a unique approach to study disturbance ecology, however, disturbances often occur in clusters across the landscape [15] (discussed by [16,17]). Thus, plots may miss clustered disturbance events, potentially over-estimating biomass and carbon stocks in forests [18] or lie in the center of a disturbance cluster and overestimate disturbance effects at the landscape level. It is obvious that heterogeneities within the landscape should be considered in studies that aim to address questions relevant to the ecosystems in which the PPs are situated.

The effect of topography is evident not only in the spatial variation of plant species distributions and vegetation structure but can also be seen from the spatial pattern in disturbance effect (e.g., fire, storms, and cyclones). For tropical cyclone disturbance, although the distance to the cyclone's eye plays a key role in the severity of forest damage across the landscape, topographic exposure is a principle factor explaining spatial patterns of cyclone-induced tree damage [19–23]. For example, cyclone-induced tree damage often varies with elevation, resulting in change of biomass and tree height along the altitudinal gradient [24,25], depending on cyclone strength (see [26]). In addition, wind exposure, which is heavily influenced by topographic position, also explains some of the spatial heterogeneity in cyclone tree damage [27,28] and vegetation structure [29] across the landscape. In general, valley and ridge vegetation are more damaged than slope vegetation [30,31], but ridgetops can be less susceptible than slopes to landslides caused by heavy rainfall [24]. Therefore, the interactions among all the topographical features lead to complex spatial patterns of damage across the landscape, which in turn affect forest resistance to cyclone disturbance [27,32]. The representativeness of PPs for the larger landscape likely varies among PPs, as they can vary greatly in physical environment (e.g., topography, soils), community composition, and agents of disturbance [2]. Thus, more PP-specific studies are needed for evaluating the representativeness of each PP to its greater landscape, before generalizations across PPs can be made.

The Fushan Forest Dynamics Plot (FFDP) is a 25-ha PP in Taiwan, and studies from the FFPD have provided important insights into forest dynamics in relation to topography, extreme climate events, and cyclone disturbance [24,33,34]. However, the representativeness of FFDP for the larger landscape has not been carefully examined. In this study, we examine how topography relates to the magnitude of typhoon-induced changes in vegetation cover between the FFDP and the broader landscape (i.e., the Fushan nature reserve). We first assess the representativeness of the FFDP for larger landscape in terms of the topography and vegetation and evaluate how the FFDP represents variation in landscape-scale typhoon disturbance effects. Then, we explore if plot representativeness varies by plot design (i.e., shape) and placement within the reserve.

## 2. Materials and Methods

### 2.1. Site, Plot, and Disturbances

The 1097 ha Fushan Experimental Forest reserve (FEF) is located in northeastern Taiwan (Figure 1), with elevation ranging from 400 to 1400 m above sea level (asl). It has an annual mean temperature of 18.2 °C, mean annual precipitation of 4270 mm, and mean relative humidity of 95% [35]. The forest is described as an old-growth sub-montane evergreen broadleaf forest [35,36]. The FEF is subject to winter monsoons, and is frequently hit by typhoons between June and October, with an average of 0.74 typhoon per year from 1951 to 2005 [37].

In 2004, the FFDP was established following the CTFS Forest-GEO standardized protocol for forest dynamic plots [36]. It is located in the western part of the FEF and ranges between 600 and 750 m asl, with approximately 84% of its area lying between 650 and 750 m asl [36].

Five typhoons were selected for this study based on the availability of high-quality Landsat images with low amounts of cloud cover. Landsat images with little cloud cover are rare because it usually rains more than 200 days per year at the FEF [37]. Thus, images with <50% cloud cover and within 30 days before and after typhoon impact were selected in order to minimize phenological changes and canopy regrowth that can take place within weeks [21,38]. The five typhoons were all of category 2 or 3 on the Saffir-Simpson scale [39] during landfall or at their nearest point to the FEF if they did not make landfall (i.e., Typhoon Aere). Moreover, for each of the five typhoons, the maximal distance between the reserve and the typhoon eye was always less than 100 km when it was nearest to the FEF so that the FEF was mostly within the radius of maximum wind of the typhoon (Figure 1).

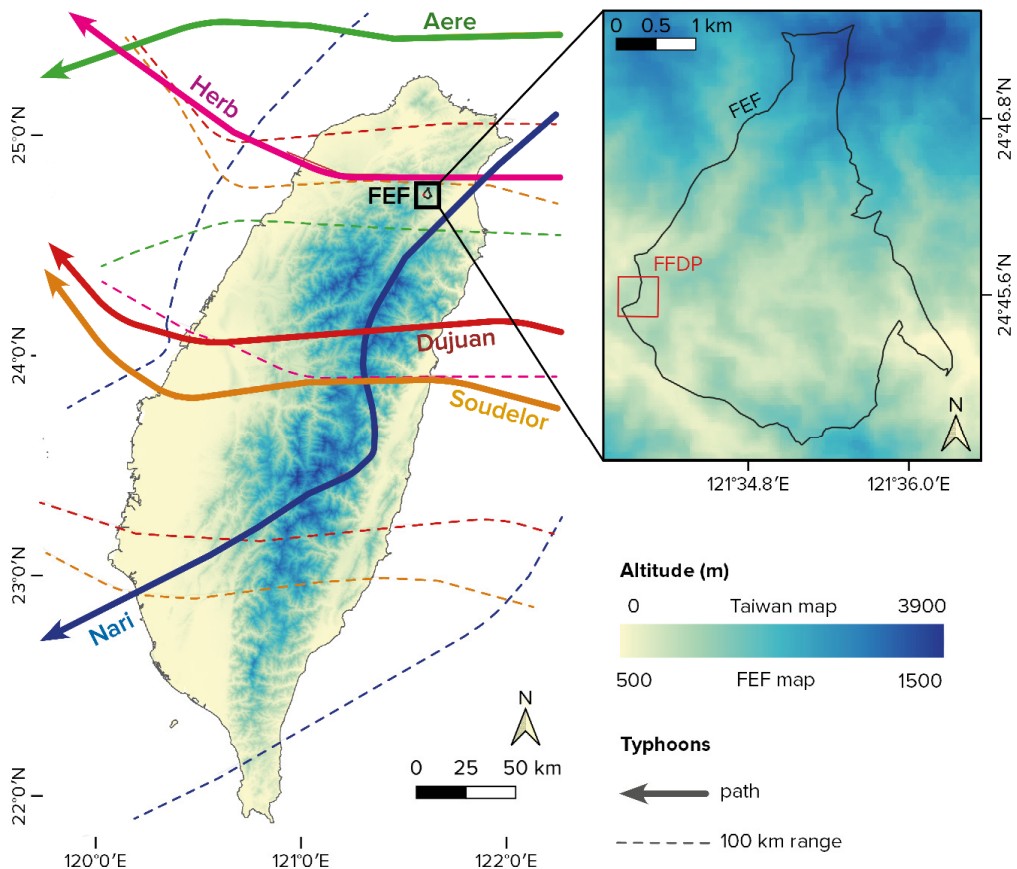

**Figure 1.** Trajectories of five typhoons analyzed in this study. Dotted lines delineate a 100 km range from the path of the typhoon eye. The location of the Fushan Experimental Forest (FEF) is labelled in the left panel and enlarged in the right panel, in which the Fushan Forest Dynamics Plot (FFDP) is also labelled. Typhoon tracks were acquired from the NOAA IBTrACS archive [40,41].

## 2.2. Data Sources

Landsats 5, 7, and 8 data were acquired from the USGS's Earthexplorer website [42] as level-2 data surface reflectance, which were atmospherically and terrain corrected. The USGS also provides a cloud mask along with level-2 data that relies on the CFmask detection algorithm [43]. The spectral features of the three sensors have been shown to be comparable, such that their data can be used in continuity to monitor forests [44,45]. A 30 m resolution DEM (digital elevation model) was acquired from the JAXA's website [46] and 30 m resolution global forest cover raster from the Global Forest Change dataset version 1.5 (GFC, [47]) was used to remove non-forested surfaces. Table 1 summarizes basic information of the images used in the study.

**Table 1.** Basic information of Landsat images of the Fushan Experimental Forest (FEF) and the Fushan Forest Dynamic Plot (FFDP). Null cells were cloud covered or non-forested surfaces. Note that because analysis-related typhoon disturbance is based on pre- and post-disturbance images within five weeks of each typhoon except for Dujuan and Soudelor, substantial cloud cover on some parts of the FEF is inevitable (see Figure S1).

| Files | Acquisition Dates | Sensors | Resolution (m) | Null Cells in FFDP (%) | Null Cells in FEF (%) |
|---|---|---|---|---|---|
| overall Fushan | 03/13/2018 | OLI | 30 | 0 | 2.5 |
| Typhoon Herb | 07/06/1996 & 08/23/1996 | TM5 | 30 | 0 | 19.4 |
| Typhoon Nari | 09/14/2001 & 10/08/2001 | TM5, ETM+ | 30 | 0 | 17.5 |
| Typhoon Aere | 07/12/2004 & 09/30/2004 | TM5 | 30 | 0 | 30.8 |
| Typhoon Soudelor | 06/09/2015 & 08/12/2015 | OLI | 30 | 1.4 | 48.8 |
| Typhoon Dujuan | 08/12/2015 & 12/02/2015 | OLI | 30 | 0 | 48.1 |

## 2.3. Pre-Processing

Landsat data was pre-processed following [48]. All reflectance rasters were topographically corrected with the *topcor* function of 'RStoolbox' package [49] in R (version 3.6) by applying a C correction on each band of JAXA's DEM. All disturbance images were partially cloudy (Table 1, see Figure S1) and clouded surfaces were removed from both pre- and post-disturbance images, and from the DEM in subsequent analysis. Visual inspection of true color composites showed that the USGS's mask removed all clouds and shadows except for Typhoon Nari, for which clouded areas were removed manually. Across all images, cloud coverage was skewed toward higher elevations (mean elevation 861 m for clouded sections against 783 m for the entire reserve). Non-forested surfaces were excluded from the analysis by masking areas with forest cover below 75% in the GFC dataset, as has been done in studies of other tropical moist forests [50,51].

## 2.4. Processing

Two vegetation indices (VIs), NDVI (Normalized Differences vegetation index, [52]) and NDII (Normalized Difference Infrared Index, [53]) were used for the overall and disturbance analysis using QGIS 3.4.2. NDVI was chosen because it is the most widely used VI for various studies around the globe [54–56] and has been shown to have a close relationship with leaf area index (LAI, [57]), and therefore forest productivity [58]. The NDII was chosen because SWIR-NIR-based indices such as NDII can track defoliation better than does NDVI [59]. The topographic position index (TPI), slope steepness and aspect were calculated with the *terrain* function from R's 'raster' package [60]. Table 2 summarizes the meaning and calculation of topographical and vegetation indices. Changes of VIs following each disturbance event, ΔVI, were calculated for each VI and disturbance events as:

$$\Delta\text{VI} = VI_{post-disturbance} - VI_{pre-disturbance}. \tag{1}$$

Numerical values of slope aspect (0 to 360) were converted into eight cardinal orientations (i.e., N, NE, E, ES, S, SW, W, and NW).

**Table 2.** Topographical variables and vegetation indices (VIs) used in this study along with their meanings and calculations based on Landsat bands and digital elevation model (DEM). R: red band; NIR: near infrared; and SWIR1: short-wave infrared.

| Index, Variable | Meaning | Calculation Method |
|---|---|---|
| **topographical** | | |
| Elevation | Altitude above sea level | - |
| Slope aspect | Surface orientation (e.g., facing North) | Orientation of each cell |
| Slope steepness | Angle to the horizontal (°) | Comparison of cell value with surrounding cells |
| Topography Position Index (TPI) | Landforms, positive values associated with ridges, negative values with valleys [61] | Comparison of a cell elevation with surrounding cells |
| **spectral** | | |
| NDVI [52] | Normalized Difference Vegetation Index. It quantifies vegetation by measuring the difference between near-infrared (which vegetation strongly reflects) and red light (which vegetation absorbs), with high values indicating dense vegetation cover. | $\frac{NIR-R}{NIR+R}$ |
| NDII [53] | Normalized Difference Infrared Index. It is sensible to vegetation water content and vegetation changes associated with cyclone disturbance [62] or defoliation [59] | $\frac{NIR-SWIR1}{NIR+SWIR1}$ |

*2.5. Analysis of FFDP-FEF Representation*

The analyses of the general FFDP-FEF representation (one Landsat image) and the representation for each typhoon disturbance (two images per event, before and after) were carried out similarly. First, mean values for VIs, ΔVIs (for typhoon disturbances only), and the three topographical variables of FEF and FFDP were statistically compared using the bootstrapped difference (via 5000 iterations) as:

$$mean\ variable_{reserve} - mean\ variable_{plot}. \tag{2}$$

There was no significant difference between the FEF and the FFDP if the 95% confidence interval of their mean difference (CI) includes 0. Slope aspects of FEF and FFDP were compared with a $\chi^2$ test. Spearman's $\rho$ was used to examine correlations between VIs, and between VIs and topography of the FFDP and FEF.

For each bootstrapped comparison of means, we calculated the coefficient of variation (CV). The post- and pre-disturbance CVs were compared between the FEF and FFDP as:

$$mean\ \left(\frac{CV_{t0}}{CV_{t1}}\right)_{reserve} - mean\ \left(\frac{CV_{t0}}{CV_{t1}}\right)_{plot}, \tag{3}$$

where $CV_{t0}$ is pre-typhoon CV and $CV_{t1}$ is post-typhoon CV. A bootstrapped comparison of means was used again to assess if typhoon-induced changes of heterogeneity of vegetation indices were different between the FFDP and the FEF.

Two thresholds were used to evaluate if a section of the FEF and FFDP was disturbed by each of the five typhoons:

$$\Delta VI < 0, \tag{4}$$

$$\Delta VI < mean_{reserve} - 0.5 * SD_{reserve}. \tag{5}$$

Although a cell with $\Delta VI < 0$ could be considered as a disturbed cell, a lower threshold (i.e., more negative $\Delta VI$) helps minimize mis-identification (e.g., due to differences in image quality). Thus,

the zone model threshold, Equation (5), was also used to define disturbed cells [63]. Spatial variation in typhoon frequency, defined as the proportion of pixels experiencing different frequencies of typhoon disturbance (0 to 5), in the FEF and FFDP were compared with a $\chi^2$ test.

The Euclidian distance measure (ED) has been used as a metric to estimate representativeness of eddy flux towers [64,65] and sampling networks [66] at the continental scale, and observation plots at the landscape scale [67]. We used ED to pinpoint well or under-represented areas of the FEF by the FFDP. The VIs (NDVI, NDII) and topographical variables (elevation, slope, TPI) were normalized (i.e., scaled to 0–1)—hence leading to a distance metric akin to the Mahalanobis distance—and then used to compute multi-dimensional distances between all FEF cells and FFDP cells. Several EDs were calculated for each reserve cell because there was more than one cell within the FFDP. Thus, the minimal values among these distances—the minimal ED (minED)—was kept for each of the reserve cells on the basis that one cell of the FFDP is more representative of a given FEF cell than the others (Figure 2). Two types of minED maps were produced: VI-based ED (a combination of NDVI and NDII) and topography-based ED (elevation, slope steepness, TPI). Correlations between topography-based and VI-based minEDs were conducted using Spearman's $\rho$.

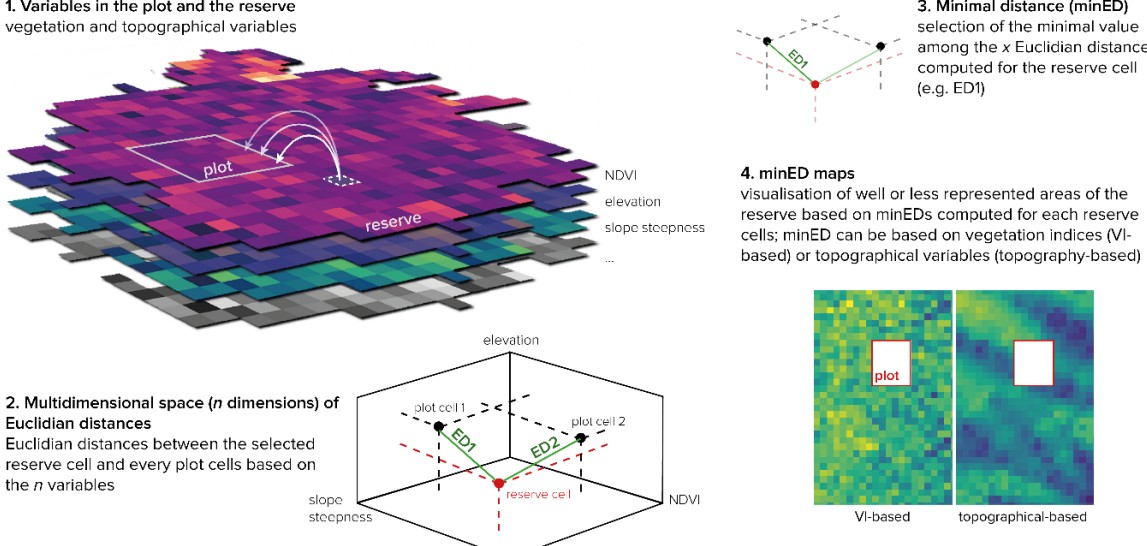

**Figure 2.** Overview of the steps for minimal Euclidian distances (minED) calculation based on vegetation indices (VIs) and topographical variables. This is a modification of regular Euclidian distance (ED) computation in which there is more than one reference cell (the plot contains *x* cells), where only the minimal value is kept. Steps leading to minED maps are described here: After grouping several data layers (step 1), Euclidian distances were computed between each plot cell and each reserve cell based on variable values standardized on a 0–1 scale (step 2). Then, step 3 involved keeping only the minimal Euclidian distance value computed for each reserve cell (among the *n* values, where *n* = the number of cells within the plot), on the basis that there was one plot cell that offered the best representation of the considered reserve cell. Finally, step 4 led to the construction of georeferenced rasters with minED values computed either with vegetation indices or topographical variables. MinEDs show how well the reserve is represented based on the inserted variables.

*2.6. Testing Alternative Forest Plot Designs*

Because vegetation cover varies with elevation and the FFDP spans a limited subset of the FEF elevational range, four alternative plot designs were tested to evaluate if increasing plot elevational range and splitting the large plot to several smaller plots, without changing the total plot area, would improve the plot's landscape representativeness. The four plot designs we tested are: one large-square plot (one 500 m × 500 m, ≥30% of FEF elevational range), four square plots (four 250 m × 250 m, ≥50% of FEF elevational range), two rectangular plots (two 250 m × 500 m transects, ≥50% of FEF elevational range),

and four rectangular plots (four 125 m × 500 m, ≥50% of FEF elevational range). The first strategy is like the FFDP, except that it covers a broader elevational range of the FEF, whereas the other three strategies use subplots that are dispersed across the landscape and cover even greater elevational ranges of the FEF. Ten plot replicates per plot design were created by randomly generating plot locations and orientations within the FEF. Plot locations were selected based on the pre-defined elevation range threshold and by limiting the number of cells with no data to <10%. Because pre-typhoon and post-typhoon images for each typhoon event were less than 30 days apart from the typhoon event (except for Dujuan), it was difficult to get disturbance-related images that lack cloud cover as it rains more than 220 days annually at the FEF [68]. After removing clouds, maintaining an elevation range >50% was not achievable. Thus, in the analysis of disturbance representativeness, the elevation range threshold was set to ≥30% of the FEF range; however, the analysis for Typhoon Aere and for the one-square strategy were excluded because clouds limited the image area available for the analyses. Ten to twelve replicates were used for each strategy. The same spatial areas were used to study the four typhoons even though they had varying cloud cover. This permitted us to compare typhoons effects in the exact same locations, and thus to maintain integrity in comparing alternative plot designs by keeping the other variables (such as slope steepness or other unmeasured factors) fixed across different typhoons. Some of the replicates of the same strategy overlap partially in areas, however subplots of each replicate do not overlap (Figure S2).

The analysis of the landscape level representativeness of the four strategies was done similar to the FFDP-FEF comparisons. Elevation was excluded from the minED calculation because this variable was used to create alternative plots. Finally, VI-based minED values for each strategy were compared through a pairwise Wilcoxon test, with Bonferroni adjustment, for each typhoon and the overall FEF. They were then ranked on a 1 to 4 scale for the overall analysis, or 1 to 3 for the disturbance analysis based on the mean minimal Euclidian distance calculated across all replicates, where the strategy with the lowest mean gets first rank.

## 3. Results

### *3.1. Overall Plot Differences between FEF and FFDP*

The difference between the FEF and FFDP was significant for both VIs as well as slope and elevation (Figure 3). Vegetation cover was greater in the FFDP, with both mean NDII and NDVI being significantly higher for pixels with the FFDP boundary (NDII = 0.25 ± 0.01, NDVI = 0.80 ± 0.009) than the greater FEF (NDII = 0.21 ± 0.03 and NDVI = 0.78 ± 0.033). Mean elevation and slope steepness were significantly higher in the reserve, whereas there was no difference for TPI between the FEF and FFDP. The proportion of the cells with different slope aspects were also significantly different between the FFDP and the FEF ($\chi^2$ = 41.86, df = 7, $p$ < 0.001, Figure S3), with a greater proportion of NW-N slope aspects and a lower proportion of SW-S aspect in the FFDP than in the FEF. The range of values represented by the FFDP in relation to the FEF was greater than 66% for the TPI, 58% for slope steepness, but only 10% for elevation, 9% for NDVI, and 23% for NDII. The interquartile range (IQR i.e., $Q_1$–$Q_3$) of the FFDP represented only 6.3% of the NDVI, 0% of the NDII, and 9.5% of the elevation, but 33.7% (slope steepness) and 81.1% (TPI) of the FEF's respective IQRs.

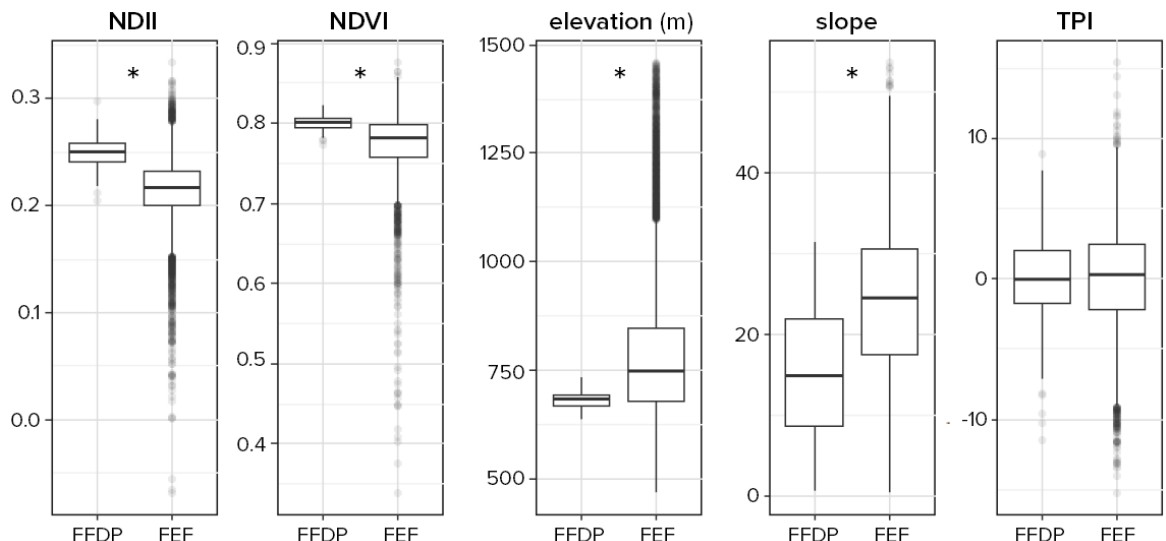

**Figure 3.** Boxplots of vegetation indices and topographical variables in the Fushan Forest Dynamics Plot (FFDP) and the Fushan Experimental Forest (FEF) based on Landsat 8 and JAXA digital elevation model (30 m spatial resolution). Significant differences between FFDP and FEF are shown with an asterisk (*), based on bootstrapped comparison on means.

In both the FFDP and FEF, the NDII and NDVI were moderately correlated ($\rho > 0.6$, $p < 0.001$; Table S1). However, the topographical variables were only weakly correlated ($\rho < 0.2$), except for TPI and elevation in the FFDP ($\rho = 0.5$, $p < 0.001$). There were no significant correlations between VIs and topographical variables except for elevation, which was negatively correlated with NDVI ($\rho_{FEF} = -0.56$, $\rho_{FFDP} = -0.34$, $p < 0.001$) and NDII ($\rho_{FEF} = -0.32$, $\rho_{FFDP} = -0.21$, $p < 0.001$) in both the FFDP and FEF.

The minimal Euclidian distance varied across the reserve (Figure 4). There was a significant, but very weak relationship between VIs-based minED and topographical variable-based minED ($\rho = 0.05$, $p < 0.001$).

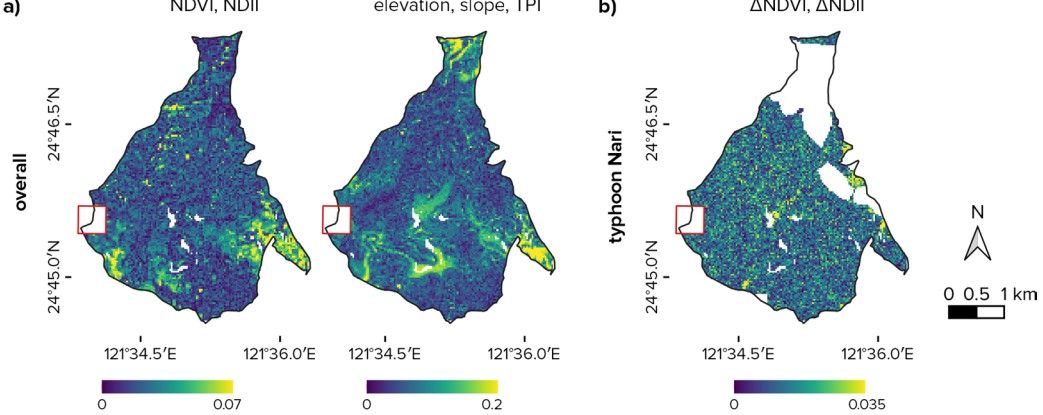

**Figure 4.** Minimal Euclidian distance between the Fushan Forest Dynamics Plot (FFDP) and the Fushan Experimental Forest (FEF) calculated from (**a**) VIs (**left**) and topographical variables (**right**); (**b**) and the disturbance effect associated with Typhoon Nari (ΔVI-based). White surface within the FEF was either obscured by cloud cover or is non-forested area. The FFDP boundary is shown via the red square.

*3.2. Disturbances Effect and Representativeness*

The mean ΔVIs were mostly negative for the five disturbances, indicating that all typhoons led to vegetation loss (Table S2). For Soudelor, Herb, and Nari, there was less vegetation loss in the FFDP than in the reserve (Figure 5; Table S2). There was, however, no significant difference for ΔNDVI with

Dujuan and ΔNDII with Nari between the FFDP and the FEF, but there was a greater vegetation loss within the FFPD than the FEF for Typhoon Aere (positive 95% CI; Table S2). Among the five typhoon disturbances, ΔNDVI of the FFDP encompassed 12.5% (Soudelor) to 47.7% (Aere) of the ΔNDVI range of the FEF, and ΔNDII of the FFDP covers 17.6% (Soudelor) to 35.4% (Dujuan) of the ΔNDII range of the FEF. However, looking at IQRs, the FFDP contained between 31.4% (Soudelor) and 75.9% (Dujuan) of the reserve's IQR for ΔNDII, and between 34.2% (Aere) and 71.0% (Dujuan) for ΔNDVI.

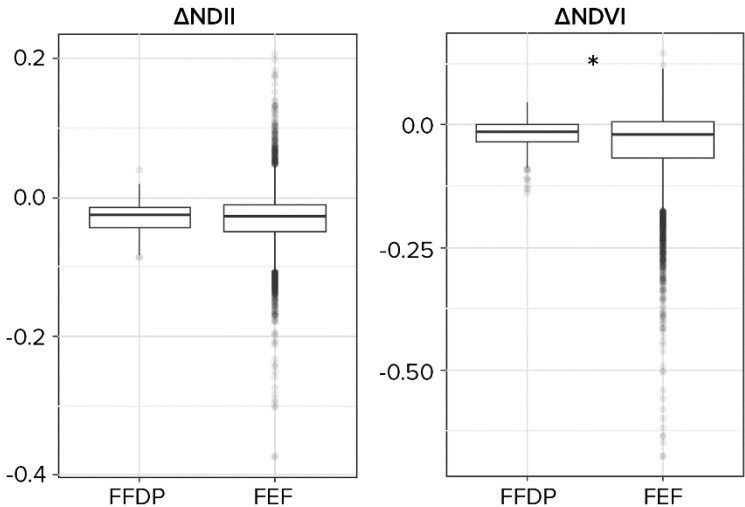

**Figure 5.** ΔNDII and ΔNDVI (calculated as the difference between post- and pre-disturbance values) for Typhoon Nari in the Fushan Forest Dynamics Plot (FFDP) and the Fushan Experimental Forest (FEF). Negative values indicate loss of vegetation. Significant differences between FFDP and FEF based on bootstrap comparison on means are shown with an asterisk (*).

ΔNDVI and ΔNDII were moderately correlated ($0.4 < \rho < 0.7$) for Typhoons Herb, Nari, and Soudelor for the FEF, were strongly correlated ($0.7 < \rho < 0.9$) for Typhoon Soudelor within the FFDP and for Typhoon Dujuan for the FEF and were very-strongly correlated ($\rho > 0.9$) for Typhoon Dujuan for both the FFDP and the FEF. However, for Typhoon Aere, the correlation between the two VIs was weak ($\rho = 0.05$ and $\rho = 0.28$ in the FFDP and the FEF, respectively). Topographical variables were weakly, if ever, correlated with ΔVIs with absolute Spearman's $\rho$ always below 0.2 (e.g., for elevation).

NDVI-based post- and pre-typhoon CV ratios were different in FFDP and FEF except for Typhoons Herb and Nari (Table S3). For both the FFDP and FEF, the ratio was below 1 (i.e., greater post- and before-typhoon CVs), indicating that typhoons caused an increase in the variation of vegetation cover. In addition, the CV ratios of the FEF were significantly closer to 1 than those of the FFDP, signifying that there was a greater difference between pre- and post-disturbance vegetation states within the FFDP than for the greater FEF (Table S3). For NDII-based CVs, significant differences between CV ratios in the FEF and FFDP were seen for Typhoons Nari, Dujuan, and Soudelor. The magnitude of change in NDII-based CVs relative to zero (i.e., no change) varied in both FFDP and FEF, with either increasing, decreasing, or equal values depending on the typhoon considered (Table S3).

The $\chi^2$ tests indicated different patterns in damage frequencies between FFDP and FEF based on ΔVI < 0 threshold ($\chi^2 = 22.48$, df = 3, $p < 0.001$ for NDVI; $\chi^2 = 16.26$, df = 4, $p = 0.003$ for NDII). For NDVI, the FFDP had more cells within the intermediate damage frequency classes (2, 3) than the FEF, whereas the proportion of higher frequency (4, 5) and no damage (0) were lower in the FFDP than in the FEF (Figure 6). The NDII also showed damage of higher intermediate frequency (3) within the FFDP than within the FEF (Table S4). Using the ΔVI < mean − 0.5 × SD threshold, the differences between the FFDP and the FEF remained ($\chi^2 = 49.00$, df = 5, $p < 0.001$ for NDVI; $\chi^2 = 33.88$, df = 5, $p < 0.001$ for NDII), even though less disturbance frequency was detected, and there was an increased proportion of undisturbed areas (class 0), especially in the FFDP.

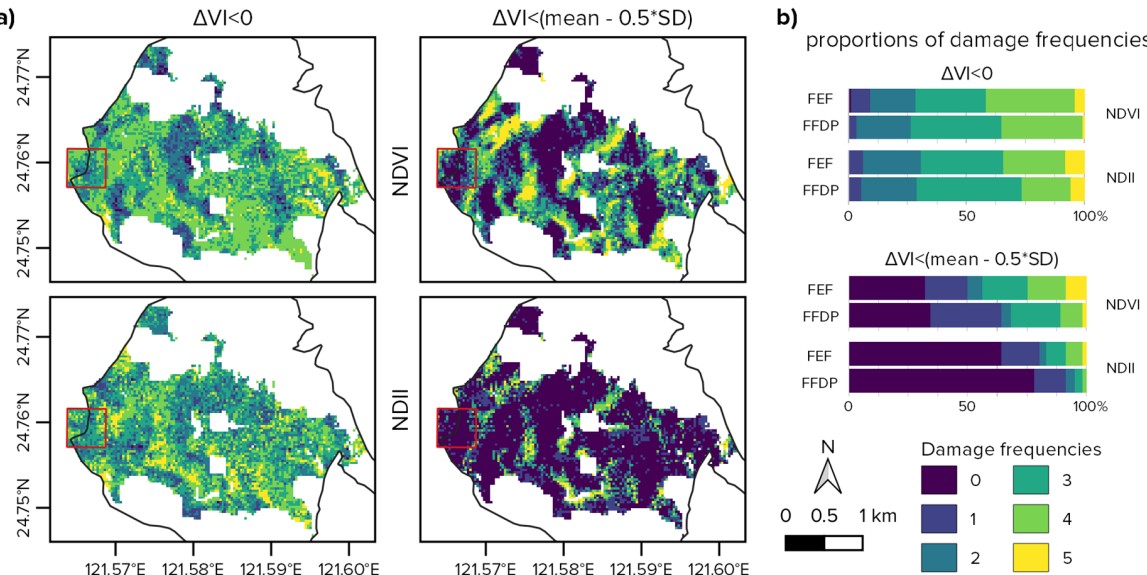

**Figure 6.** (**a**) Spatial representations of disturbance occurrences for NDVI (**top panels**) and NDII (**bottom panels**) for 5 typhoons (listed in Table 1) based on $\Delta$VI < 0 (**left**) and $\Delta$VI < mean $-$ 0.5 $\times$ SD (**right**) thresholds. Damage class ranks from 0, being undisturbed for all five typhoons, to 5, having been disturbed by all typhoons. The Fushan Forest Dynamics Plot (FFDP) is shown in red, while the Fushan Experimental Forest (FEF) boundary is in black. White areas were either obscured by clouds or are outside FEF-FFDP limits. (**b**) Proportions of each damage frequency class within the FEF and the FFDP for both vegetation indices and based on the two thresholds ($\Delta$VI < 0, **top**; $\Delta$VI < mean $-$ 0.5 $\times$ SD, **bottom**). The percentage of cells for each damage class for both thresholds are shown in Table S4.

### 3.3. Alternative Plot Designs

The replicates of the same plot design had significantly different vegetation and topography from the FEF (Table S5). The reserve had greater VIs (denser vegetation) than half of the alternative plot designs (e.g., 22 out of 40 replicates based on NDVI; Table S5). For topographical variables, most of the alternative designs led to significant differences with the FEF for slope steepness (24 out of 40 replicates; Table S5). However, the aspects proportions of the alternative plots were significantly different between the FEF and every alternative plot design replicate ($p < 0.01$), except one replicate of the four-rectangular plot design ($\chi^2$ test, $\chi^2$ = 12.33, df = 7, $p$ = 0.09). Despite numerous differences between the FEF and the alternative plots for NDVI and NDII, one plot of the two-rectangular plot design strategy, one of the four-squares strategy, one of the four-rectangular plot, and one of the one-square plot strategy showed no significant differences in NDVI and NDII from those of the FEF (Table S5), despite having much greater elevation ranges.

The topography-based minED of alternative plot designs was significantly greater than that of the FFDP for all but two replicates (four-rectangles design, plot 8 and four-squares design, plot 3). Figure 7 shows the minED values calculated from the FFDP and the four-rectangles design replicates (ranked first in plot designs overall comparison, Table 3), whereas other plot designs are shown in Figure S4. However, VI-based minEDs showed the opposite trend, as most alternative plots led to improved representativeness of NDVI and NDII (i.e., for 30 out of 40 replicates the VI-based minED was lower than that of the FFDP, Figure 7 and Figure S4).

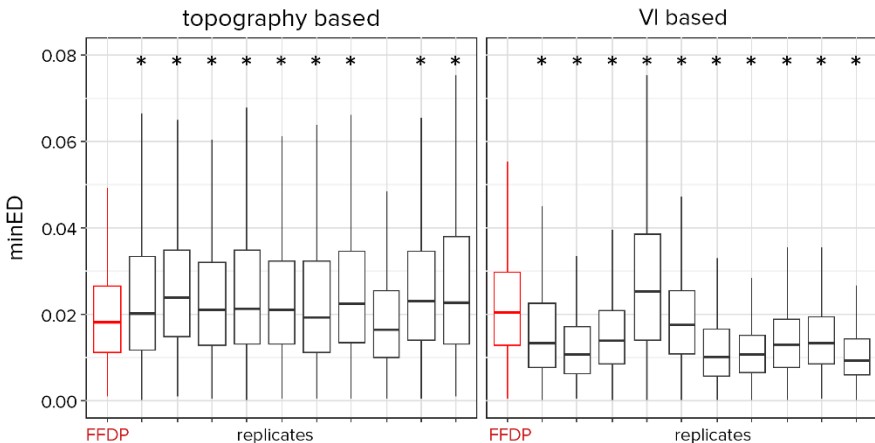

**Figure 7.** Minimal Euclidian distances (minED) measured between the Fushan Experimental Forest (FEF) and the Fushan Forest Dynamics Plot (FFDP) for the four-rectangles plot design strategy (all designs shown in Figure S4). Two types of minED were calculated: based on topographical variables (**left panel**), and based on vegetation indices (VI, **right panel**). Significant differences between the minED calculated for the FFDP and the alternative plots shown with an asterisk (*, see Table S5).

For Typhoon Nari, the comparison between alternative plot designs and the greater FEF showed that most of the alternative plot designs were less exposed to disturbances (i.e., $\Delta VI_{alternative} > \Delta VI_{FEF}$, Table S6) than is the FEF. Higher damages within the alternative plot designs were found for 1 of 11 replicates for the four-rectangular plot design, 4 out of 12 for the four-squares plot strategy, and 5 out of 10 replicates for the two-rectangular plot strategy. However, in the four-rectangular plot design, three replicates were not significantly different from the reserve for $\Delta NDII$ (Table S6).

The FFDP had a lower mean minED than most of the alternative plot design strategies for Typhoons Dujuan, Herb, and Soudelor (Table S7). However, it was not the case for Typhoon Nari, wherein several alternative plot designs had smaller minEDs than the FFDP. See Figure 8 for Typhoons Herb and Nari for the four-rectangular design (other typhoons and plot designs shown in Figure S5).

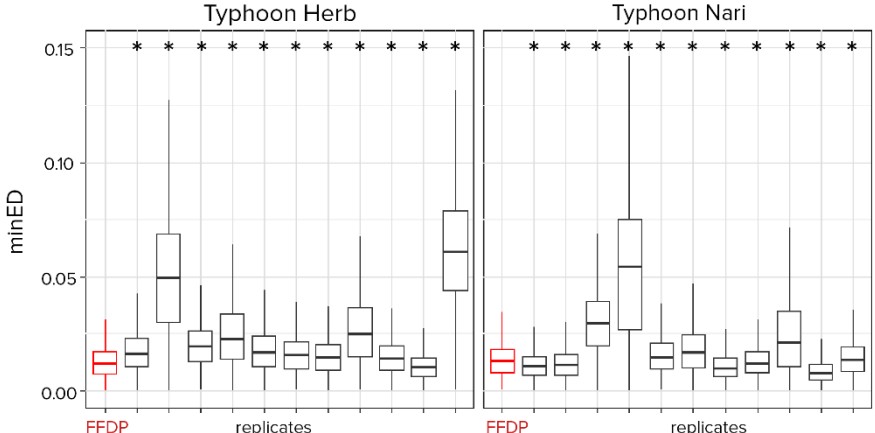

**Figure 8.** Minimal Euclidian distances (minED) for changes of vegetation indices ($\Delta VIs$) associated with Typhoons Herb and Nari between the Fushan Forest Dynamics Plot (FFDP) and the four-rectangular plot design strategy within the Fushan Experimental Forest (FEF). Data for other plot designs and typhoons are shown in Figure S5. Significant differences between the FFDP and the alternative plots are shown with an asterisk (*) based on 95% CIs (see Table S7).

The mean minED of plot designs strategies were significantly different in each typhoon event as well as for the overall vegetation cover (Table S8). The two-rectangular plot design had the best ranking (i.e., smallest mean minED, meaning best FEF representation) among the three alternative

plot design strategies across the four typhoon events, but it did not consistently have the smallest minED for all of the four events (Table 3). For the overall vegetation cover, the one-square plot design replicates led to the highest mean minEDs (i.e., least representativeness) among the four alternative plot design strategies (Table 3).

**Table 3.** Ranks of mean minimal Euclidian distances (ED) of four alternative plot design strategies based on vegetation indices for representativeness of difference effects (i.e., ΔVIs) and of the overall vegetation cover for the Fushan Experimental Forest. Significant differences between means of different plot design strategies were detected in all cases with a Wilcoxon test ($p < 0.05$, with Bonferroni adjustment), $p$ values shown in Table S8.

| Strategies | Typhoon Disturbances | | | | | Overall |
| --- | --- | --- | --- | --- | --- | --- |
| | Dujuan | Herb | Nari | Soudelor | Total | |
| two rectangles | 3 | 1 | 1 | 2 | 7 | 2 |
| four squares | 2 | 2 | 3 | 1 | 8 | 3 |
| four rectangles | 1 | 3 | 2 | 3 | 9 | 1 |
| one large square | - | - | - | - | - | 4 |

## 4. Discussion

### 4.1. Overall Representativeness

The greater vegetation cover as indicated by both VIs, the overall gentler slopes, and lower altitude of the FFDP in relation to the FEF, and the significant differences in slope aspect between the FFDP and the FEF (Figure 3) all suggest that the plot does not adequately represent the greater landscape, both in terms of vegetation and topographic heterogeneity. Given the much smaller area of the FFDP (25 ha) compared to the FEF (1097 ha) and the rough mountain topography of the reserve, it is not surprising that the FFDP cannot adequately represent the topography of FEF. However, although the FFDP covers only 10% of the elevation range of the FEF, it covers more than 50% of the landform variation (TPI) and slope steepness of the FEF. The high coverage of the landform variability and slope steepness, however, does not lead to a high degree representativeness in vegetation as the range of VIs of the FFDP covers only 9% of the NDVI and 23% of the NDII ranges of the FEF (Figure 3). The lack of significant correlation between the topographical variables and vegetation indices within both the FFDP and the FEF (Table S1) explains that the moderately high topographical representativeness of FFDP for the FEF does not lead to high representativeness of vegetation indices.

Elevation is the only topographical factor examined that had any significant correlation with VIs. The negative relationship between elevation and VIs can partially explain the greater VIs in FFDP than the FEF because the FFDP has a considerably lower mean elevation than the FEF. Many environmental factors important to plant growth, such as temperature and precipitation vary with elevation such that vegetation cover is likely to vary with elevation. Notably, the correlation between elevation and VIs are stronger for the FEF than the FFDP. Elevation is likely a more important factor for structuring vegetation cover at the broader landscape than within the FFDP, which has smaller elevational range. Many studies illustrate that tropical forest tree species diversity decreases with elevation [25,69,70], including those from northern Taiwan [71], the narrow elevation range within the FFDP probably limits its representativeness of the landscape tree species richness by overestimating its diversity and missing species that only grow at higher elevations [71]. It has also been reported that forest NDVI is closely related to tree diversity (see review by [72]), thus the difference in NDVI between the FFDP and the FEF suggests that the FFDP could have different species composition from the greater FEF.

Higher values and narrower ranges of NDVI were positively related to LAI [73] and NDII, which were, in turn, positively related to vegetation water content, suggesting that the FFDP overestimates LAI (NDVI, [73]) and forest water status (NDII, [74]) at the landscape scale,

but underestimating their variability. Thus, the FFDP is unlikely a completely representative sample for primary productivity and other important environmental processes at the landscape (i.e., FEF) scale.

### 4.2. Typhoons Damages Intensity in the Plot and the Reserve

The significant differences of typhoon-induced ΔVIs between the FFDP and the FEF (Table S4) suggest that the FFDP cannot adequately represent the disturbance effects of the greater FEF. The low representativeness of typhoon disturbance within the FFDP is also evident from the differences in the proportion of the cells affected with respect to typhoon frequency between the FFDP and the FEF (Figure 6). For both VIs, less than 9% of the cells of the FEF were hit by all of the five typhoons examined (Table S4), and the small size of the FFDP relative to the FEF certainly limit its ability to capture cells with high typhoon frequencies Figure 6). Effects of topography on the cyclone damage distribution within forested landscapes are widespread across regions affected by tropical cyclones [23,75,76]. In this study, elevation was the only topographical variable that correlated with typhoon damages for three of the five studied typhoons, although relationships were weak ($\rho < 0.2$). This suggests that slope steepness and TPI, which are relatively well represented by the FFDP (Figure 3), have limited influence on forest sensitivity to wind disturbances at the 30 m resolution in the reserve and in the permanent plot. This contrasts with the report of lesser forest damage by disturbance in flatter surfaces of the FFDP in a ground survey of aboveground biomass [24]. More studies are required to examine if such differences are related to differences in the survey methods (i.e., remote sensing vs. ground-based approaches), typhoon variability, or scale. Notably, FFDP representativeness for typhoon disturbance effect is considerably better than its representativeness of overall vegetation cover, as it represents between 30% and 75.9% of the FEF IQRs associated with ΔVIs across the five typhoons, but only less than 7% of the FEF IQRs for VIs. Thus, despite covering a small fraction of the vegetation variability and elevational range of the greater landscape, the FFDP provides a relatively good representation of disturbance effects occurring at the landscape scale. In addition, our results did not suggest any over- or under-exposure to cyclone damage within the FFDP relative to the FEF over the long term, because differences in ΔVIs are not consistent across typhoons. Moreover, even though statistically significant, the differences between FFDP and FEF may have limited ecological meaning as the ΔVIs values are often very close in absolute value (e.g., a difference of 0.01 of mean NDVI for typhoon Herb; Table S2).

### 4.3. Vegetation Cover and Topographical Representativeness with Alternative Strategies

Across all strategies, most of the alternative plot designs improved the representativeness of vegetation cover in comparison with the FFDP (lower minED values), although the representativeness of topographical parameters, slope steepness, and TPI decreased (Figure 7). However, the elevation range of the plot increased from less than 10% to over 30% of the FEF range due to the alternative plot designs. The lack of consistency between vegetation and topographical representativeness could be explained by the lack of correlation between TPI, slope steepness, and the VIs. Moreover, the one-square strategy led to fewer replicates with improved vegetation representativeness than the other strategies with spatially separate subplots (Table S5), likely because its elevational range is narrower than the other strategies as elevation was significantly correlated with VIs. Nevertheless, the elevation range may not be the only factor explaining increased VI representation as dispersed subplots across the landscape may also provide a more representative sample of the vegetation diversity than a single large plot in landscapes with high spatial heterogeneity such as Fushan. Indeed, it has been reported that spreading multiple plots across a forest landscape led to increased representativeness of plot biomass and biodiversity patchiness in comparison with a single large plot [77].

### 4.4. Disturbances Representativeness with Alternative Strategies

The evaluation of the different plot design strategies with respect to disturbance effect led to an unexpected trend. Despite its smaller elevational range, the FFDP can be more representative of the FEF (i.e., smaller minED) in terms of typhoon effects than alternative plot designs with greater

elevational ranges. This was observed for Typhoons Dujuan, Herb, and Soudelor, which is 75% of cases (Typhoon Aere was not included in the analysis of alternative plot design strategies). On the other hand, several alternative plot designs showed improved ΔVI representativeness for Typhoon Nari (Figure 8). Since the same alternative plot placements were used to study all disturbances, the difference between Nari and other events may be the product of less cloud cover in the Landsat images used to study the effects of this typhoon. Clouds obstructed almost 50% of the FEF for Typhoons Dujuan and Soudelor, but less than 20% for Typhoon Nari. Clouds were mostly located at higher elevations and elevation has weak but significant correlations with ΔVIs; therefore, the representativeness of disturbance effects across the FEF within the alternative plot design strategies (e.g., minED values) may vary among typhoons because of varying cloud cover and elevational ranges of the observable area. Nevertheless, for Typhoon Nari, alternative plots had ΔVIs that were significantly different from the reserve and most remained underexposed just like the FFDP (Table S6). Thus, a complete representation of typhoon disturbances may not be attainable by using the current plot nor by alternative plot design strategies based on 25 ha surface but wider elevational ranges.

*4.5. Comparison of Strategies*

The comparison of the alternative plot design strategies based on minED for both the overall vegetation cover (four strategies, including one-square) and disturbance effects (three strategies) did not show an overall improvement in plot representativeness with respect to the FFDP (Table S7). However, the strategies that used multiple subplots ranked better than one large square and two rectangular plot strategies, which is in agreement with the observations of a previously published study by Salk and colleagues [77]. Furthermore, in contrast to the result from a simulative study by Reese and colleagues [78], we do not find greater representativeness of rectangular shaped plots (i.e., transects) relative to squares suggesting that the possibility of rectangular quadrats to cover a wider range of vegetation heterogeneity is not universal.

## 5. Conclusions

Several studies have explored the complexity of the landscape representation of forest plots [15,18,77,79]. Within a tropical forest reserve in Taiwan, this study shows that VI-derived forest parameters and typhoon damage at the landscape scale are not adequately represented by the current permanent forest dynamics plot. Across the five studied tropical cyclones, the FFDP does not display a trend for more, or less damage, or for more, or less post-disturbance heterogeneity than the FEF. However, an underestimation of forest damage was found for Typhoon Nari, the typhoon analyzed with the lowest cloud coverage. Hence, upscaling of damages observed at FFDP to the landscape scale needs to be done with care and depends on image representativeness of the landscape. Examining alternative plot design strategies led to different landscape representativeness, which varied with disturbances, and no single plot design strategy performed significantly better than the current FFDP. We conclude that although the FFDP does not represent the FEF completely, it is no less representative than any possible alternative plots that are likely more difficult to conduct ground surveys in, as they cover a much wider elevational range and steeper slopes. Further exploration of alternative plotting strategies is necessary, possibly by using additional environmental parameters (e.g., soils) and other thresholds. Moreover, the representativeness of the tested alternative plot designs is not known for other sites exposed to different disturbance regimes, thus further research on the representativeness of disturbance effects within other large permanent plots is needed. Focus should be extended to multiple sites from different biomes (e.g., temperate, lowland forests), disturbances regimes (rare to frequent) by using hyper- and multispectral imagery, as well as LiDAR data that can track ecosystem processes over time in relation to disturbance. The use of a standardized approach for representativeness estimation (e.g., computation of Mahalanobis distances based on the same variables) could allow for cross-sites comparisons. An improved knowledge of how findings from large permanent plots scale up to

the landscape level effects of cyclones disturbances is valuable, particularly in a time when cyclone disturbance regimes are changing and will likely alter forest dynamics [80–83].

**Supplementary Materials:** The following are available online at http://www.mdpi.com/2072-4292/12/4/660/s1, Figure S1: Cloud covers in the FEF and FFDP for 5 typhoon disturbances, Figure S2: Location of plots created following the four alternative strategies for the overall and disturbance analysis, Figure S3: Proportion of each slope aspect in the FEF and the FFDP, Figure S4: Minimal Euclidian distances measured between the FEF and the FFDP for the four alternative plot designs based on vegetation indices and topographical variables, Figure S5: minED for changes of vegetation indices associated with four typhoons between the FFDP and the three alternative plot design strategies within the FEF. Table S1: Overall analysis Spearman's $\rho$ for correlation between vegetation indices and topographical variables for plot and reserve, Table S2: Mean ΔVIs for the five typhoons in the FFDP and FEF, Table S3: Coefficient of variation of four variables before and after disturbances, with bootstrapped comparison on means 95% CIs, Table S4: Percentage of cells included in each damage sum class based on the two thresholds and two vegetation indices for the FFDP and FEF, Table S5: The 95% CIs from comparisons between alternative plots and FEF for VIs and topographical variables, and comparison of minED of FFDP and alternative plots, Table S6: 95% CIs from comparisons of ΔVIs between alternative plots and the FEF for Typhoon Nari, Table S7: 95% CIs from comparisons ΔVI-based minED between alternative plots and the FFDP for four typhoons, Table S8: Wilcoxon tests *p* values for the comparisons of mean minED between each plot designs for all typhoons (except Aere) and the overall vegetation analysis.

**Author Contributions:** Conceptualization, T.-C.L. and J.P.; methodology, T.-C.L. and J.P.; software, J.P.; validation, T.-C.L., J.P. and J.A.H.; formal analysis, J.P.; investigation, J.P. and T.-C.L.; resources, J.P.; data curation, J.P. and J.A.H.; writing—original draft preparation, J.P.; writing—review and editing, J.A.H. and T.C.L.; visualization, J.P. and T.-C.L.; supervision, T.-C.L.; project administration, T.-C.L.; funding acquisition, T.-C.L. All authors have read and agreed to the published version of the manuscript.

**Funding:** This research is supported by grants from the Ministry of Science and Technology (MOST 107-2313-B-003-001-MY3).

**Acknowledgments:** We thank Yi-Ching Lin, Pei-Jen Lee Shanner, Andy Jan, and Mao-Ning Tuanmu for comments on earlier versions of the manuscript.

**Conflicts of Interest:** The authors declare no conflict of interest. The funders had no role in the design of the study; in the collection, analyses, or interpretation of data; in the writing of the manuscript, or in the decision to publish the results.

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
