# Peer review of "Landscape Representation by a Permanent Forest Plot and Alternative Plot Designs in a Typhoon Hotspot, Fushan, Taiwan"

_remotesensing, doi:10.3390/rs12040660_

Round 1
Reviewer 1 Report
The contribution is innovative and interesting. It brings new knowledge for evaluation of representativeness of research / monitoring plots. It creatively uses, integrates and complements existing knowledge from current and relevant solutions published so far.
The aim to propose an approach to assess the representativeness of the monitoring plot using the Mahalobis distance in the space of relevant factors and to verify the suitability of the solution on the example of forest damage by tropical storms is actual, ambitious and demanding. Reversing the logic of the method offers the possibility of designing an alternative selection design of research / monitoring plots. The proposed method based on the relevant staggering procedures is well documented and explained. I have no comments on the first part of the task. In the second part I consider it problematic to limit the choice of the location of alternative locations of the monitoring areas by the possibility of obtaining input data for the evaluation (cloudiness of satellite scenes). On the other hand, I realize the importance of such an approach in terms of the integrity of the overall approach. I suggest the authors try to eliminate (bypass) or at least minimize the impact of this fact.
From a technical point of view, the paper is written in a comprehensible style, and its structural breakdown is also generally suitable. Quality and understanding of principles supporting are also included tables and visualizations including supplementary data. The proposed method and its results are adequately presented. The work contains all the essential parts required in the scientific publication - a balanced analysis of the issue and the identification of unsolved, respectively, otherwise solved problems, clear goal definition. I consider the source data used, the process of their processing as well as the statistical apparatus to be appropriate, qualifyied and justified. Innovative is the use of Mahalobis distance calculations for this purpose. There are some reserves to assess the quality of the results achieved and to compare them with the results of other authors in the field, together with recommendations for generalization. From a terminological point of view, I do not think it is right to talk about Euklideans but Mahalobis distances.
The correctness of the procedures used seems to be undeniable. The study is technically sound, using proven and widely accepted approaches, data, methods, tools, etc. It is also reproducible. The formulation and commenting of scientific results has some reserves due to the lack of subsequent comparison with the results of other authors.
The results can be very interesting once the conclusions have been finalized and demonstrated.
The work brings new knowledge.
The text is comprehensible and written in a suitable style. But I do not feel competent to assess the level of language.
Author Response
The contribution is innovative and interesting. It brings new knowledge for evaluation of representativeness of research / monitoring plots. It creatively uses, integrates and complements existing knowledge from current and relevant solutions published so far.
Reply: We appreciate the time that Reviewer 1 has contributed to revewing our manuscript. The overall positive comments on our manuscript were helpful in improving it.
The aim to propose an approach to assess the representativeness of the monitoring plot using the Mahalobis distance in the space of relevant factors and to verify the suitability of the solution on the example of forest damage by tropical storms is actual, ambitious and demanding. Reversing the logic of the method offers the possibility of designing an alternative selection design of research / monitoring plots. The proposed method based on the relevant staggering procedures is well documented and explained. I have no comments on the first part of the task. In the second part I consider it problematic to limit the choice of the location of alternative locations of the monitoring areas by the possibility of obtaining input data for the evaluation (cloudiness of satellite scenes). On the other hand, I realize the importance of such an approach in terms of the integrity of the overall approach. I suggest the authors try to eliminate (bypass) or at least minimize the impact of this fact.
Reply: We agree with the reviewer that it is ideal to eliminate or at least minimize the impact of cloudiness of satellite images. However, “as it rains more than 220 days annually at the FEF [68]” (Lines 226), and the pre- and post-typhoons images are within 30 days of the typhoon events, obtaining images with low levels of cloud covers was not possible for all typhoons. We had to limit the choice of the locations for the simulated alternative plot designs because “the same spatial areas were used to study the four typhoons even though they had varying cloud cover. This permitted us to compare typhoons effects in the exact same locations, and thus to maintain integrity in comparing alternative plot designs by keeping the other variables (such as slope steepness or other unmeasured factors) fixed across different typhoons.” (Lines 230-234) The phrases/sentences in quotations are added to the revised manuscript.
Chang, C.-T.; Wang, L.-J.; Huang, J.-C.; Liu, C.-P.; Wang, C.-P.; Lin, N.-H.; Wang, L.; Lin, T.-C. Precipitation controls on nutrient budgets in subtropical and tropical forests and the implications under changing climate. Adv. Water Resour. 2017, 103, 44-50. doi:10.1016/j.advwatres.2017.02.013.
From a technical point of view, the paper is written in a comprehensible style, and its structural breakdown is also generally suitable. Quality and understanding of principles supporting are also included tables and visualizations including supplementary data. The proposed method and its results are adequately presented. The work contains all the essential parts required in the scientific publication - a balanced analysis of the issue and the identification of unsolved, respectively, otherwise solved problems, clear goal definition. I consider the source data used, the process of their processing as well as the statistical apparatus to be appropriate, qualifyied and justified. Innovative is the use of Mahalobis distance calculations for this purpose. There are some reserves to assess the quality of the results achieved and to compare them with the results of other authors in the field, together with recommendations for generalization. From a terminological point of view, I do not think it is right to talk about Euklideans but Mahalobis distances.
Reply: We appreciate the positive comment about the writing of the manuscript. As for the terminological issue, to make our results comparable to other studies that assessed representativeness we calculated Euclidean distance. Based on our knowledge and search on the web, to our understanding the Mahalobis distance is similar to the Euclidian distance if all axis were standardized. We modified the following sentence to address this issue:
“We used ED to pinpoint well or under-represented areas of the FEF by the FFDP. The VIs (NDVI, NDII) and topographical variables (elevation, slope, TPI) were normalized (i.e., scaled to 0-1) – hence leading to a distance metric akin to the Mahalanobis distance – and then used to compute multi-dimensional distances between all FEF cells and FFDP cells.” (Lines 188-191)
The correctness of the procedures used seems to be undeniable. The study is technically sound, using proven and widely accepted approaches, data, methods, tools, etc. It is also reproducible. The formulation and commenting of scientific results has some reserves due to the lack of subsequent comparison with the results of other authors.
The results can be very interesting once the conclusions have been finalized and demonstrated.
The work brings new knowledge.
The text is comprehensible and written in a suitable style. But I do not feel competent to assess the level of language.
Reply: We appreciate the overall positive comments. Regarding the reserves, due to the lack of subsequent comparison with the results of previously published studies, we have attempted to do so. We point the reviewer to two specific points from the discussion: “However the strategies that used multiple subplots ranked better than one large square and two rectangular plot strategies, which is in agreement with the observations of a previously published study by Salk and colleagues [76]. Furthermore, in contrast to the result from a simulative study by Reese and colleagues [77].” (Lines 490-493)
Nonetheless, we acknowledge the need for more comparisons with other studies and highlighted this in the (revised) conclusion by suggesting “The use of a standardized approach for representativeness estimation (e.g., computation of Mahalanobis distances based on the same variables) could allow for cross-sites comparisons.” (Lines 516-518)

Reviewer 2 Report
An interesting and well-presented manuscript overall.
It is much preferred to see some more quantitative results shown and displayed in the abstract. As it is it is quite qualitative.
The upper right part of Figure 1 should have some more description as to location context as it is difficult to interpret as part of the rest of the figure.
In some of the multi-part figures it would be useful to label them a, b, c, etc. and describe each in more detail in the figure caption.
Was any specific consideration made for comparing the NDVI values between the different sensors? Literature may suffice.
Within Figure 2 please specify the descriptions of the different sub parts 1, 2, 3, 4, within the figure caption.
Some references appear to need to be re-configured for proper display.
Overall an interesting and thoughtful manuscript with some fairly well-prepared and helpful graphics.
Author Response
An interesting and well-presented manuscript overall.
Reply: We appreciate the positive comments to our manuscript.
It is much preferred to see some more quantitative results shown and displayed in the abstract. As it is it is quite qualitative.
Reply: Following the comment, we added some quantitative results to the abstract while removing some of the qualitative descriptions to keep roughly the same number of words as in the original manuscript. Specifically, the following quantitative results are added to the revision.
“Results showed that the FFDP neither represents landscape elevational range (< 10%) nor vegetation cover (< 7% of the interquartile range, IQR).” (Lines 26-28)
“In addition, the ΔVIs were of the same magnitudes in the plots and the reserve, and the plot covered 30% to 75.9% of IQRs of the reserve ΔVIs.” (Lines 30-31)
“Based on the comparison of mean Euclidian distances, with two rectangular plots having the smallest distance than four square or four rectangular plots of the same overall area.” (Lines 34-35)
The upper right part of Figure 1 should have some more description as to location context as it is difficult to interpret as part of the rest of the figure.
Reply: We modified figure with lines zooming out to the right panel so that it is clear that the upper right part of the figure is a zoom in of the FEF labeled in the large figure.
In some of the multi-part figures it would be useful to label them a, b, c, etc. and describe each in more detail in the figure caption.
Reply: We added a, b to label sub-figures of figures 4 and 6. (Lines 274-275 for Figure 4; Line 313 and Lines 318-320 for Figure 6)
Was any specific consideration made for comparing the NDVI values between the different sensors? Literature may suffice.
Reply: We added the following to describe our consideration on this issue. (Lines 123-125)
“The spectral features of the three sensors have been shown to be comparable, such that their data can be used in continuity to monitor forests [44,45].”
44. She, X.; Zhang, L.; Cen, Y.; Wu, T.; Huang, C.; Baig, M.H.A. Comparison of the continuity of vegetation indices derived from Landsat 8 OLI and Landsat 7 ETM+ data among different vegetation types. Remote Sens. 2015, 7, 13485-13506.
45. Vogelmann, J.E.; Gallant, A.L.; Shi, H.; Zhu, Z. Perspectives on monitoring gradual change across the continuity of Landsat sensors using time-series data. Remote Sens. Environ. 2016, 185, 258-270. doi:10.1016/j.rse.2016.02.060.
Within Figure 2 please specify the descriptions of the different sub parts 1, 2, 3, 4, within the figure caption.
Reply: We added the following to specify the description of the sub-parts. (Lines 203-210)
“Steps leading to minED maps are described here: After grouping several data layers (step 1), Euclidian distances were computed between each plot cell and each reserve cell based on variable values standardized on a 0-1 scale (step 2). Then, step 3 involved keeping only the minimal Euclidian distance value computed for each reserve cell (among the n values, where n = the number of cells within the plot), on the basis that there was one plot cell that offered the best representation of the considered reserve cell. Finally, step 4 led to the construction of georeferenced rasters with minED values computed either with vegetation indices or topographical variables. MinEDs show how well the reserve is represented based on the inserted variables.”
Some references appear to need to be re-configured for proper display.
Reply: We have corrected the two places where incurred referenced in page 10. (Line 333 and Line 342)
Overall an interesting and thoughtful manuscript with some fairly well-prepared and helpful graphics.
Reply: We appreciate the positive comment.

Reviewer 3 Report
In this manuscript the authors present a strategy for assessing the representation of forest plots within a larger environment in a Taiwan reserve based on the analysis of mainly satellite measurements. Different vegetation and topographic variables are used for comparing plot vs. reserve dynamics by considering also the effects of 5 different typhoon disturbances. Similar studies are based on ground and/or meteorological data and models for assessing cyclone effects. Here, a rather detailed statistical analysis and comparison is performed with the combined use of remote sensing variables and various alternative scenarios.
The overall quality of the manuscript is high with a clear structure and a detailed presentation of methods and results. The results can provide insight to the forest plot's relationship with the overall reserve, since they are based on extensive computations of different statistical metrics. I do not have any substantial comments, after going through all the tables and checking the consistency with the in-text discussion of the results.
A minor question/clarification: In page 6 where the minED map generation is discussed, the VI-based ED is based on the minimum distance calculated form a single index or a combined minimum from both?
Author Response
In this manuscript the authors present a strategy for assessing the representation of forest plots within a larger environment in a Taiwan reserve based on the analysis of mainly satellite measurements. Different vegetation and topographic variables are used for comparing plot vs. reserve dynamics by considering also the effects of 5 different typhoon disturbances. Similar studies are based on ground and/or meteorological data and models for assessing cyclone effects. Here, a rather detailed statistical analysis and comparison is performed with the combined use of remote sensing variables and various alternative scenarios.
Reply: We appreciate the positive comments about our manuscript.
The overall quality of the manuscript is high with a clear structure and a detailed presentation of methods and results. The results can provide insight to the forest plot's relationship with the overall reserve, since they are based on extensive computations of different statistical metrics. I do not have any substantial comments, after going through all the tables and checking the consistency with the in-text discussion of the results.
Reply: We appreciate the positive comments.
A minor question/clarification: In page 6 where the minED map generation is discussed, the VI-based ED is based on the minimum distance calculated form a single index or a combined minimum from both?
Reply: We added the following to make it clear that it is a combined minimum from both. “Two types of minED maps were produced: VI-based ED (a combination of NDVI and NDII) …..” (Lines 195-196)
